# SWE-Refactor: A Repository-Aware Benchmark for Evaluating LLMs on Real-World Code Refactoring

## Abstract

Recent advances in Large Language Models (LLMs) have garnered significant attention for their applications in software engineering tasks. Among these tasks, code refactoring has its own unique challenges. Unlike code generation, refactoring requires precise changes that preserve program behavior while improving structure, making automated evaluation difficult. Existing refactoring benchmarks suffer from three key limitations: (1) they often focus on atomic refactoring types while missing more complex ones; (2) they contain noisy data with entangled, unrelated code changes, making it difficult to study LLM's true refactoring capability accurately; and (3) they lack code repository and structural information to support realistic evaluations. To address these issues, we propose *SWE-Refactor*, a new benchmark for LLM-based code refactoring. *SWE-Refactor* contains 1,099 real-world, pure refactorings collected from 18 real-world Java projects. Each refactoring instance is verified through compilation, test execution, and automated refactoring detection tools to ensure correctness. Unlike prior benchmarks, *SWE-Refactor* covers both atomic and compound refactoring types (single and multiple code changes). It includes rich repository-level data (e.g., method callers and callees, class hierarchies), as well as configuration details like test coverage and build settings. We evaluate nine widely used LLMs on *SWE-Refactor*, including GPT-4o-mini, DeepSeek-V3, and CodeLLaMa. DeepSeek-V3 achieves the best performance with 457 successful refactorings (41.58%), followed by GPT-4o-mini with 438 (39.85%). DeepSeek-V3 performs particularly well on *Extract Method*, completing 301 cases, while GPT-4o-mini demonstrates stronger performance on more complex refactoring types, such as *Move Method* and *Extract and Move Method*. Furthermore, we find that adding retrieval context via few-shot examples and using a multi-agent workflow significantly improve performance, with the multi-agent approach achieving the highest success rate. We release *SWE-Refactor* and all evaluation results to support future research on LLM-based code refactoring.

## 1 Introduction

In software engineering, code refactoring is a process of improving the structure of existing code without changing its behavior (Fowler, 1999). This practice is essential for maintaining software systems by improving code quality, enhancing reusability, and ensuring adaptability to changing requirements (Murphy-Hill et al., 2011). Unlike coding, code refactoring typically involves analyzing existing code to identify code segments for improvement, understanding its structure and dependencies, and then making precise changes without altering its behavior. For example, a common refactoring operation is *Extract Method* (Fowler, 1999; Murphy-Hill et al., 2011; Tsantalis et al., 2020), where a developer identifies a portion of a long method that can operate independently and extracts it into a separate method, making the original method shorter, more readable, and reusable.

In recent years, Large Language Models (LLMs) have been widely applied across various software engineering tasks due to strong abilities in code understanding and reasoning (Lin et al., 2024; Jin et al., 2023; Alshahwan et al., 2024; Qin et al., 2024). Among these tasks, code generation has attracted significant attention (Lin et al., 2024; Jiang et al., 2024; Ishibashi & Nishimura, 2024), where LLMs generate code from natural language descriptions or specifications. In contrast, code

Table 1: The comparison between existing benchmarks and *SWE-Refactor*. **Compound refactoring** means there can be multiple code transformations. **Pure refactoring** indicates commits without unrelated changes. **Developer-written GT** refers to the ground truth refactored code being written by *original project developers*. **Test availability** shows whether test cases are provided to verify correctness. **Automated construction** indicates whether the benchmark was built entirely via an automated pipeline.

| Benchmark | Code Distribution | | Compound | Pure | Developer- | Test | Automated |
|---|---|---|---|---|---|---|---|
| | # Repo | # Sample | Refactoring | Refactoring | Written GT | Availability | Construction |
| ref-Dataset (Liu et al., 2025) | 20 | 180 | ✗ | ✓ | ✓ | ✗ | ✗ |
| community corpus (Pomian et al., 2024) | 5 | 122 | ✗ | ✗ | ✓ | ✗ | ✗ |
| extended corpus (Pomian et al., 2024) | 12 | 1,752 | ✗ | ✗ | ✓ | ✗ | ✓ |
| RefactorBench (Gautam et al., 2025) | 9 | 100 | ✗ | ✗ | ✗ | ✓ | ✗ |
| *SWE-Refactor* | 18 | 1,099 | ✓ | ✓ | ✓ | ✓ | ✓ |

refactoring poses a different challenge, *requiring a deep understanding of existing code semantics and repository structures, and making precise changes that preserve the original behavior while improving code structures*. This creates unique challenges, as LLMs must precisely determine what to change while preserving the functional behaviors. Moreover, evaluating refactoring capabilities requires realistic settings and codebases, since real-world code introduces complex design patterns, dependency chains, and language features that are rarely captured in synthetic examples.

To assist with these challenges, mainstream integrated development environments (IDEs) such as IntelliJ IDEA (JetBrains, 2024a), PyCharm (JetBrains, 2024b), and Eclipse (Foundation, 2024) have introduced semi-automated refactoring tools. These tools can help perform low-level code changes but still rely heavily on developers to understand the code and make key decisions. To further reduce manual effort and enhance automation, recent studies have investigated the use of LLMs for code refactoring tasks (Pomian et al., 2024; Shirafuji et al., 2023; White et al., 2024; Xu et al., 2025), and several benchmarks have been proposed to evaluate model performance. However, these benchmarks often have one or more of these four key limitations, as summarized in Table 1.

❶ **Consider Only Atomic Refactoring Types**. Existing refactoring benchmarks often focus on a limited set of *atomic refactoring types (i.e., a single code transformation)*. Figure 1 shows an example where an *Extract Method* appears *as part of* a **compound refactoring (i.e., multiple code transformations)**. As shown in Table 1, the *community corpus* (Pomian et al., 2024) and *extended corpus* (Pomian et al., 2024), used to evaluate EM-Assist (an IntelliJ plugin), focus exclusively on one atomic (*Extract Method*) refactoring. Similarly, the *ref-Dataset* proposed by Liu et al. (2025) supports 9 atomic refactoring types, including *Extract Method* and *Extract Variable*, but lacks support for more complex, compound refactorings such as *Extract And Move Method*. *RefactorBench* (Gautam et al., 2025) also focuses on a limited set of 7 atomic refactoring types, including *Move Class*, *Rename Class*, *Move Method*, and *Rename Method*. Definitions for each refactoring type are provided in Appendix C. **In short, none of the existing benchmarks support compound refactorings**.

❷ **Noisy Benchmark Data.** Existing refactoring benchmarks often contain code changes that are not purely refactoring. This occurs because refactoring activities are mostly driven by changes in requirements (such as new features and bug fixes), and less driven by solely code smell resolution (Silva et al., 2016). However, impure changes make it hard to determine whether the LLM-generated code aligns with the intended refactoring. If the reference solution contains both refactorings and other functional changes, it becomes unclear which types of changes the model is expected to generate. This ambiguity reduces the effectiveness of benchmarks for evaluating code refactoring. As shown in Table 1, among all existing benchmarks, only *ref-Dataset* (Liu et al., 2025) contains pure refactorings, where the authors manually removed the refactoring from the modified code to recreate the original version. This method works for simple refactorings, such as *Rename Method*, but is hard to apply to more complex cases that involve multiple files, like *Move Method*, due to manual overheads.

❸ **Insufficient Support for Repository-Level Analysis and Automated Verification**. Existing refactoring benchmarks are not designed to evaluate LLM's capability in repository-level tasks. They typically include only basic elements such as task descriptions, code before and after refactoring, and lack the additional repository-level information (e.g., method callers and callees, class hierarchies,

and inheritance relationships) required for more advanced refactoring or repository-level analyses. Moreover, most benchmarks do not provide tests for automated verification. Among all existing benchmarks, only *RefactorBench* (Gautam et al., 2025) includes associated tests.

❹ **Lack of Automated Construction.** Many existing benchmarks are not automatically constructed, requiring manual effort in various stages such as preparing pre-refactoring code or writing ground truth and test cases. Specifically, *ref-Dataset* (Liu et al., 2025) manually reverts code changes to reconstruct pre-refactoring code, which is both time-consuming and error-prone. *RefactorBench* manually constructs, with the help of LLM, both the ground truth refactored code and the corresponding test cases. These manual steps make the benchmarks difficult to scale and maintain. Some changes even go beyond refactoring, such as modifying repository logic, which shifts the focus away from behavior-preserving code refactorings.

Existing software engineering benchmarks also suffer from a significant imbalance in programming languages. A recent study by Cao et al. (2024) shows that 95.6% of the latest benchmarks are built exclusively on Python (e.g., SWE-bench (Jimenez et al., 2024), HumanEval (Chen et al., 2021), MBPP (Austin et al., 2021), and RefactorBench (Gautam et al., 2025)), limiting the diversity and representativeness of evaluation. To bridge this gap and address the above-mentioned challenges, we introduce *SWE-Refactor*, a benchmark for evaluating LLMs' code refactoring capabilities on Java projects. Java is one of the most widely used programming languages in the world, ranking among the top in both the TIOBE index (TIOBE Software BV, 2025) and the Stack Overflow developer survey (Stack Overflow, 2024). Java's statically typed and syntactically structured grammar also results in well-defined refactoring patterns, allowing for more precise and accurate refactoring benchmarking. By focusing on Java, *SWE-Refactor* broadens evaluation beyond the current Python-centric landscape and reflects the languages used in large-scale enterprise and open-source systems.

*SWE-Refactor* consists of 1,099 pure refactorings extracted from 18 widely used Java projects, complementing existing benchmarks (e.g., *RefactorBench*) that predominantly focus on Python. ❶ **In addition to atomic, it also covers compound refactoring types**, including three atomic types—*Extract Method*, *Move Method*, and *Inline Method*—as well as three compound types—*Extract and Move Method*, *Move and Inline Method*, and *Move and Rename Method*. ❷ *SWE-Refactor* **eliminates noises and includes only pure refactoring**. To ensure the purity of refactoring, we use abstract syntax tree (AST)-based refactoring detection tools that are shown to have great precision (98%) and recall (91%) (Tsantalis et al., 2018; 2020; Nouri, 2023) to extract and select only pure refactoring from a large number of real-world refactoring code commits. ❸ *SWE-Refactor* **provides comprehensive repository-level information**. In addition to the basic information (code before refactoring, developer-written refactored code, and refactoring type), *SWE-Refactor* provides rich repository-level and structure information, including project structure, class body, caller and callee of method, build configuration details, and test coverage information. ❹ *SWE-Refactor* **ensures automated and reproducible data collection**. *SWE-Refactor* fully automates the extraction of pure refactoring data from real-world projects, avoiding the need for manual annotation or LLM-generated code. All ground-truth refactored code is directly derived from project repositories. This ensures scalability and future benchmark expansion. ❺ **High quality and executable refactoring.** *SWE-Refactor* extracts developer-written refactorings from real-world projects with diverse application domains, allowing it to better reflect the capabilities of LLMs in realistic software engineering scenarios. To ensure the reliability of the benchmark, we perform multi-stage verification: (i) AST-based static analysis to confirm that each commit contains only the targeted refactoring type and no unrelated code changes, (ii) compilation and execution of the full test suite to confirm behavioral equivalence, and (iii) manual checks on a subset of instances to prevent false positives from automated tools. We retain only those refactorings that pass all verification steps, ensuring that *SWE-Refactor* contains high-quality, executable, and behavior-preserving examples. Details on the project selection and the distribution of refactorings are provided in Appendix D.

We evaluate 9 widely used LLMs (GPT-4o-mini (OpenAI, 2023), GPT-3.5 (OpenAI, 2023), DeepSeek V3 (DeepSeek-AI et al., 2024), Qwen2.5 Coder (Hui et al., 2024), DeepSeek Coder (Guo et al., 2024), and CodeLLaMa (Rozière et al., 2023)) on our proposed *SWE-Refactor* benchmark. We evaluate the refactored code along two dimensions: functional correctness and human-likeness. For functional correctness, we assess the code using 1) compilation success and test pass rate, and 2) AST-Based Refactoring Verification, which verifies that the expected refactoring has indeed occurred in the modified code. For human-likeness, we employ the *CodeBLEU* metric (Ren et al., 2020) to measure the difference. We find that the performance of large general-purpose LLMs is significantly

better than that of open-source LLMs. DeepSeek V3 achieves the best results across all metrics, successfully refactored 457 out of 1,099 cases (41.58%). GPT-4o-mini ranks second, with 438 successful refactorings (39.85%). Furthermore, the performance of LLMs on different refactoring types is significantly different. DeepSeek V3 leads in *Extract Method*, completing 301 cases, while GPT-4o-mini shows the strongest performance on compound refactoring types, such as *Extract And Move Method*.

Overall, our contributions in this work are threefold:

- We introduce *SWE-Refactor*, a benchmark constructed from developer-written commits that contain only refactorings and no other functionality changes. It is designed to comprehensively evaluate LLM's capabilities on both atomic and compound refactoring tasks.
- We design a fully automated four-step pipeline to construct *SWE-Refactor*, which extracts real refactorings, filters out impure ones, collects relevant structural information, and verifies functional correctness through compilation and test execution.
- We conduct an extensive evaluation of 9 popular LLMs on *SWE-Refactor* and perform a fine-grained analysis of their performance across different refactoring types, highlighting their strengths and limitations.

## 2 RELATED WORK

**Refactoring Benchmarks.** *RefactorBench* (Gautam et al., 2025) is a Python-based benchmark for evaluating the effectiveness of LLM agents on code refactoring. Unlike *SWE-Refactor* that leverages developer-written refactorings mined from real commits, RefactorBench relies on LLMs to identify refactoring opportunities, which can introduce model-specific biases into the benchmark. Moreover, *SWE-Refactor* captures the complex real-world software design, including overridden methods, generics, exception handling, and inheritance hierarchies that are often missing in synthetic data. RefactorBench's ground truth solutions are also manually written by the authors, who may not have in-depth knowledge of the project. *ref-Dataset* (Liu et al., 2025) includes 100 pure atomic refactorings from real Java projects. The *community corpus* provides 122 Extract Method refactorings from five older Java projects. The *extended corpus* (Pomian et al., 2024) expands this to 1,752 Extract Method instances. However, each of the benchmarks has its own limitation, as shown in Table 1. Our benchmark, *SWE-Refactor*, is automatically built from 18 modern Java projects, covering both atomic and compound refactorings. All ground truth refactored code and test cases are written by the original project developers. The benchmark supports automated evaluation and ensures both structural and behavioral correctness through compilation and full test verification.

**LLMs-based Code Refactoring.** Recent works have explored various techniques to enhance LLM performance in refactoring tasks, including prompt clarity (AlOmar et al., 2024), structured prompting (White et al., 2024), and few-shot learning (Shirafuji et al., 2023). Hybrid approaches that combine LLMs with rule-based systems have also shown improved results (Zhang et al., 2024). Several works directly prompt models like GPT-4 to perform refactorings (DePalma et al., 2024; Poldrack et al., 2023), confirming the feasibility of using LLMs for this task. In addition, practical tools such as *EM-Assist* (Pomian et al., 2024) and the Context-Enhanced Framework (Gao et al., 2024) demonstrate how LLMs can be integrated into automated refactoring workflows. Our benchmark can serve as a basis for future work in this area by providing a standardized and real-world dataset to evaluate and compare refactoring capabilities of LLMs across both atomic and compound transformations.

## 3 SWE-REFACTOR

### 3.1 OVERVIEW

Figure 1 shows a data sample of *SWE-Refactor*. Each sample in *SWE-Refactor* contains 6 components.

❶ **Target Method**: The original method code before refactoring. ❷ **Refactoring Type**: The specific refactoring operation applied to the target method. For example, the data sample in Figure 1 illustrates an *Extract and Move Method* refactoring, where a block of code is first extracted into a separate method and then moved to a more appropriate class. ❸ **Repository and Code Structure**: Structural information of the target method at the repository, class, and method levels. Repository-level details

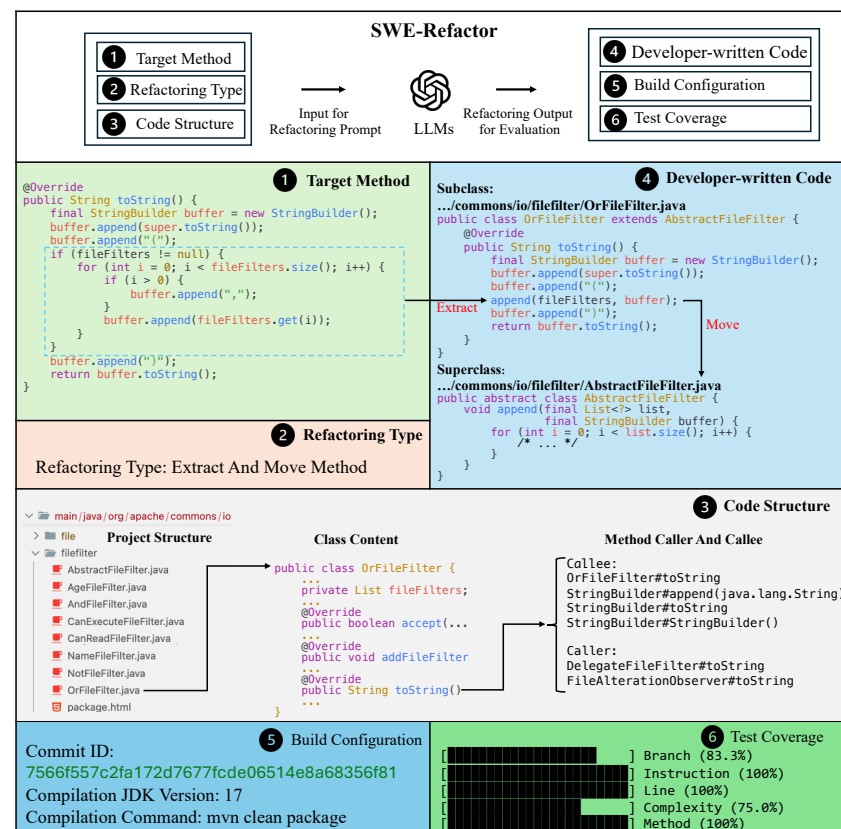

Figure 1: An overview of the data in *SWE-Refactor*.

include the overall project structure and the full paths to all source Java files in the repository. Class-level details include the source code of the entire class and hierarchy (i.e., parent and child relationships). Method-level information includes method's callers and callees. ❹ **Developer-Written Code**: The target method refactored by project developers, serving as a reference for evaluating the quality of LLM-generated refactored code. ❺ **Build Configuration**: Compilation-related information necessary for building the project after refactoring. This includes the commit ID, the compatible JDK version, and the specific build commands. ❻ **Test Coverage**: Coverage data showing how the target method is exercised by the test suite. Comparing coverage before and after refactoring helps verify whether the refactoring preserves the program's functional behavior.

## 3.2 TASK AND VERIFICATION METRICS

As illustrated in Figure 1, *SWE-Refactor* is designed to evaluate the performance of Large Language Models (LLMs) in real-world code refactoring. Given a target method, a specific refactoring type, and relevant repository and source code information, *SWE-Refactor* helps assess how effectively LLMs can generate correct and human-like refactored code. To evaluate refactoring quality from multiple perspectives, we employ three evaluation metrics: compilation and test success, AST-Based Refactoring Verification, and *CodeBLEU*.

❶ **Compilation and Test success (Functional Verification)**. *SWE-Refactor* integrates the LLM-generated refactored code into the project, then compiles the project and runs its test suites. This step verifies the functional correctness, ensuring the generated refactored code does not break the build or introduce unexpected issues.

❷ **AST-Based Refactoring Verification (Refactoring Verification)**. While compilation and test success reflect functional correctness, they do not guarantee that the intended refactoring has been applied and may risk overfitting to the test suite. Due to potential hallucination issues in LLMs (Huang et al., 2023b), they may generate code that passes tests but deviates from the intended refactoring. To address this, we use *RefactoringMiner* (Tsantalis et al., 2020), an Abstract Syntax Tree (AST)

and rule-based static code analysis tool for detecting Java code refactorings, to verify whether the LLM-generated code contains the intended refactoring and to ensure the code contains no other functionality changes. *RefactoringMiner* has excellent performance at identifying refactorings within complex and mixed-purpose commits, achieving an average precision of 99% and recall of 94% in detecting refactoring (Tsantalis et al., 2020).

❸ **CodeBLEU (Human-Likeness Verfication)**. Finally, even when the code is functional and the refactoring is correct, it may still differ in quality or readability from the refactored code written by a human developer. Therefore, we include *CodeBLEU* (Ren et al., 2020) to assess the human-likeness of the generated code. *CodeBLEU* is a code-specific evaluation metric that compares the textual, structural, and semantic similarities between two code snippets. By considering multiple dimensions, it provides a more accurate assessment of how closely the generated code matches what a human developer would write.

## 3.3 AUTOMATED BENCHMARK CONSTRUCTION PIPELINE

Figure 2 presents the automated pipeline of building *SWE-Refactor*. Unlike *RefactorBench* (Gautam et al., 2025), which synthesizes refactoring examples using LLMs, our dataset is built from real-world refactorings written by humans, identified through traditional static code and AST analysis. This design choice ensures the benchmark is free from LLM-induced hallucinations or bias. To construct *SWE-Refactor*, we design a four-step automated pipeline:

**Step 1: Mine Refactorings via Static Analysis**. We leverage AST-based refactoring detection tools to extract commits that contain refactorings from GitHub repositories. *RefactoringMiner* is an AST- and rule-based tool that demonstrates high accuracy in refactoring detection. In addition to identifying refactoring types, we apply static code analysis to analyze the Java files. For each detected refactoring instance, we analyze the code and extract the detailed location information, including the commit hash, the affected Java files, and the specific line numbers within the file. This information is also stored in *SWE-Refactor* as part of our released dataset. Based on this information, we further build the ASTs of the modified Java files. Then, we traverse the ASTs to extract Method Level and Class Level information for the refactoring instance, including the source code before and after the developer's refactoring changes, and the method and class signatures.

**Step 2: Curate Pure and Targeted Refactoring Types.** After extracting all commits containing refactorings, we use AST-based pure refactoring detection tools to curate high-quality instances by filtering out impure changes (e.g., bug fixes) and retaining only the six refactoring types studied in this work. *PurityChecker* (Nouri, 2023) extends *RefactoringMiner* with specialized AST analysis to identify pure method-level refactorings, with an average precision of 95% and recall of 88%. It starts by identifying refactorings in a commit and comparing the code before and after the refactoring. During this process, *PurityChecker* analyzes how original statements are changed—specifically, which statements were moved, modified, or replaced as part of the refactoring. It then checks whether these changes follow predefined purity rules.

**Step 3: Enrich Refactoring Changes with Multi-Level Code Information.** *RefactoringMiner* analyzes refactorings within individual Java files and does not support cross-file analysis or method invocation. Hence, we further use the Eclipse Java Development Tools (Eclipse JDT) (Eclipse Foundation, 2024) to extract structural information at the repository, class, and method levels. Eclipse JDT is a static analysis tool that provides access to the ASTs and type bindings of Java projects. For each refactoring instance, we identify the modified Java files and collect additional source files within the same software package. We implement static analysis tools to analyze these files and construct ASTs with resolved types and method references. By traversing the ASTs, we extract the repository structure, the source code of the entire class and its hierarchy, and caller-callee relationships.

**Step 4: Verify Compilation and Test Coverage.** For each refactoring, we develop a script to compile the project and verify its correctness. To determine the appropriate JDK version, we attempt compilation using multiple JDKs. We then execute the test suite with JaCoCo (Jacoco, 2009) to collect code coverage information and exclude commits where the refactored code is not exercised by any test. Finally, we verify the existence of target classes involved in *Move Method*, *Extract and Move Method*, and *Move and Inline Method* refactorings. This step was necessary because the *Move Method* operation may move a method to newly created classes, and it is difficult for LLMs to predict the newly created classes.

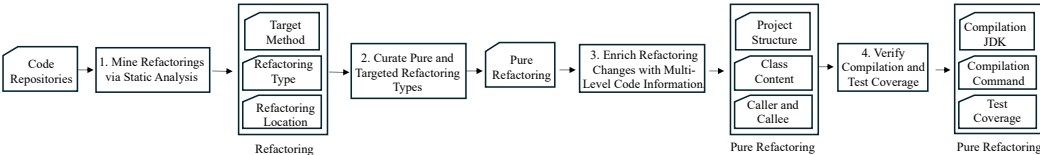

Figure 2: Our Automated Pipeline to Construct *SWE-Refactor*.

Table 2: Evaluation of 9 LLMs on *SWE-Refactor*. The table presents the number of refactorings to perform, compile-and-test success rates, refactoring correctness verified by AST-Based refactoring detection tools (AST-Based RF Verification), and code similarity to human-written refactorings (*Code BLEU*). **Successful Refactoring** refers to the number of refactorings that compile, pass tests, and are verified by AST-Based refactoring detection tools. We report the average *Code BLEU* score and total counts for the other metrics.

| Model | Size | Compile&Test Success | AST-Based RF Verification | Code BLEU | Successful Refactoring |
|---|---|---|---|---|---|
| gpt-4o-mini | N/A | 537 (48.86%) | 636 (57.87%) | 0.547 | 438 (39.85%) |
| gpt-3.5-turbo | N/A | 199 (18.11%) | 142 (12.92%) | 0.536 | 82 (7.46%) |
| DeepSeek-V3 | N/A | **554 (50.41%)** | **674 (61.33%)** | **0.584** | **457 (41.58%)** |
| Qwen2.5 Coder | 14B | 22 (2.00%) | 101 (9.19%) | 0.428 | 7 (0.64%) |
| Qwen2.5 Coder | 7B | 20 (1.82%) | 142 (12.92%) | 0.582 | 6 (0.55%) |
| DeepSeek Coder | 16B | 23 (2.09%) | 101 (9.19%) | 0.549 | 3 (0.27%) |
| DeepSeek Coder | 6.7B | 31 (2.82%) | 70 (6.37%) | 0.442 | 7 (0.64%) |
| CodeLLaMa | 13B | 14 (1.27%) | 15 (1.36%) | 0.558 | 1 (0.09%) |
| CodeLLaMa | 7B | 41 (3.73%) | 48 (4.37%) | 0.502 | 12 (1.10%) |

## 4 EXPERIMENT

In this section, we evaluate 9 popular LLMs on *SWE-Refactor*, and analyze their effectiveness across different refactoring types, prompting strategies, and multi-agent workflows. They cover general LLMs (i.e., gpt-4o-mini-2024-07-18 (OpenAI, 2023), gpt-3.5-turbo-01-25 (OpenAI, 2023), and DeepSeek-V3 (DeepSeek-AI et al., 2024)) and Code LLMs (Qwen2.5 Coder-{7b, 14b} (Hui et al., 2024), DeepSeek Coder-{6.7B, 16B} (Guo et al., 2024), and CodeLLaMa-{7B,13B} (Rozière et al., 2023)). General LLMs are accessed via official APIs, while Code LLMs are deployed on a cluster with 4 NVIDIA A100 GPUs (40GB each).

### 4.1 LLMs' PERFORMANCE ON *SWE-Refactor*

We evaluate 1,099 pure refactorings from the *SWE-Refactor* using the three metrics defined in Section 3.3: Compilation and Test Success, AST-Based Refactoring Verification, and *CodeBLEU*. A refactoring is considered successful if it passes both Compilation&Tests and AST-Based Refactoring Verification. For consistency, we design a standardized prompt template containing four components: (1) a task description of the refactoring, (2) the target method, (3) repository-level context such as class source and caller–callee relations, and (4) a natural language instruction specifying the expected transformation. The detailed prompt template is provided in Appendix E. As shown in Table 2, DeepSeek-V3 achieves the best overall performance with 457 successful refactorings (41.58%), followed by GPT-4o-mini with 438 (39.85%). General-purpose LLMs substantially outperform open-source code LLMs, reflecting their stronger capabilities in code understanding. Among the open-source models, CodeLLaMa-7B performs best with 12 successes (1.10%), while the 13B variant performs worse, likely due to its Python-focused pre-training (Chai et al., 2025), which highlights the importance of having a non-Python benchmark.

### 4.2 PERFORMANCE ACROSS REFACTORING TYPES

To better understand how LLMs perform on different kinds of refactorings, we analyze their effectiveness across the six refactoring types studied in *SWE-Refactor*: three atomic types (*Extract Method*, *Move Method*, *Inline Method*) and three compound types (*Extract and Move Method*, *Move and Inline Method*, and *Move and Rename Method*). For each refactoring type, we compute the

Table 3: Performance of LLMs across six refactoring types. **EM** = Extract Method, **IM** = Inline Method, **MM** = Move Method, **RM** = Rename Method. Values in parentheses indicate the total number of instances per refactoring type collected in the *SWE-Refactor*.

| Model | Size | Successful Refactoring | EM (441) | IM (71) | MM (410) | EM + MM (142) | MM + RM (21) | MM + IM (14) |
|---|---|---|---|---|---|---|---|---|
| gpt-4o-mini | N/A | 438 | 259 | **53** | 92 | **33** | **1** | 0 |
| gpt-3.5-turbo | N/A | 82 | 48 | 9 | 23 | 2 | 0 | 0 |
| DeepSeek-V3 | N/A | **457** | **301** | 50 | 76 | 30 | 0 | 0 |
| Qwen2.5 Coder | 14B | 7 | 2 | 5 | 0 | 0 | 0 | 0 |
| Qwen2.5 Coder | 7B | 6 | 5 | 1 | 0 | 0 | 0 | 0 |
| DeepSeek Coder | 16B | 3 | 1 | 1 | 0 | 1 | 0 | 0 |
| DeepSeek Coder | 6.7B | 7 | 6 | 1 | 0 | 0 | 0 | 0 |
| CodeLLaMa | 13B | 1 | 1 | 0 | 0 | 0 | 0 | 0 |
| CodeLLaMa | 7B | 12 | 12 | 0 | 0 | 0 | 0 | 0 |

success rate based on Compilation and Test Success and AST-Based Refactoring Verification. This analysis helps reveal whether certain LLMs are more effective at atomic refactorings compared to compound ones, and whether some types pose more challenges for current models. Table 3 shows that DeepSeek-V3 achieves the strongest specialization on *Extract Method* with 301 successes, while GPT-4o-mini exhibits broader generalization, particularly in cross-file tasks such as *Move Method* (92) and *Extract+Move* (33). Open-source models (Qwen2.5, DeepSeek Coder, and CodeLLaMa) succeed mainly only on a few *Extract Method* instances.

Overall, the table highlights a clear trend: current LLMs remain effective on local atomic edits but perform poorly on cross-file and compound transformations. These tasks thus represent critical benchmarks for advancing LLMs' reasoning ability over structured software artifacts.

### 4.3 IMPACT OF CONTEXT AUGMENTATION AND MULTI-AGENT WORKFLOWS

To examine the effect of context augmentation and multi-agent reasoning, we extend beyond simple prompting on *SWE-Refactor* using two techniques. We apply Retrieval-Augmented Generation (RAG) to provide additional context via retrieved refactoring examples, and a multi-agent workflow that iteratively refines the outputs. We evaluate both techniques using `gpt-4o-mini`, chosen for its strong performance on complex refactorings and tool support.

RAG provides more context to LLMs through relevant few-shot examples, aiming to improve the accuracy and relevance of the generated code

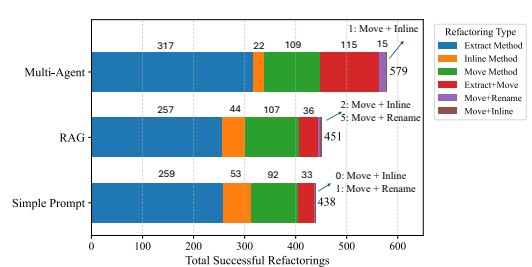

Figure 3: Comparison of successful refactorings.

(He et al., 2024; Shirafuji et al., 2023). Our RAG implementation uses a retrieval database of 905 pure refactoring instances drawn from the *Refactoring Oracle Dataset* (Tsantalis et al., 2020), which has no overlap with the data in *SWE-Refactor* (construction details in Appendix F). The multi-agent workflow strengthens the reasoning and validation abilities of LLMs (Huang et al., 2023a). We define two roles: a *Developer Agent*, which generates refactored code given context, and a *Reviewer Agent*, which critiques the output and provides iterative feedback. This design enables multi-turn refinement while mitigating common reasoning failures (Appendix G).

As shown in Figure 3, the Multi-Agent strategy achieves the highest overall success (579 refactorings), outperforming RAG (451) and Simple Prompting (438). While all three perform similarly on *Extract Method*, the Multi-Agent workflow shows clear advantages on more complex refactoring, completing 109 *Move Method* and 115 *Extract+Move* cases, far exceeding RAG (107, 36) and Simple Prompt (92, 33). These improvements likely stem from iterative reasoning and feedback between agents.

## 4.4 Scalability to SOTA Models and Agentic Scaffolding

To further assess the limits of *SWE-Refactor*, we extended our evaluation to stronger models: (1) `GPT-4o` Hurst et al. (2024) as the base model in our multi-agent workflow, and (2) an agentic scaffolding setup using OpenAI Codex (`GPT-5.1-Codex`) OpenAI (2025).

**Performance of GPT-4o.** We replaced the base model with `GPT-4o` in our agent approach. Table 4 summarizes the full benchmark results. `GPT-4o` achieves 675 out of 1,099 successful refactorings (61.4%), a clear improvement over `gpt-4o-mini` (52.7%). The largest gains appear in navigation-intensive refactoring types, such as *Move Method* and *Move And Inline Method*, suggesting that `GPT-4o`'s stronger reasoning and repository-navigation capabilities help the agent locate the correct files and apply the required edits more reliably.

Table 4: GPT-4o results on full *SWE-Refactor* (1,099 instances).

| Model | Total(Success) | EM | IM | MM | EM+MM | MM+RM | MM+IM |
|---|---|---|---|---|---|---|---|
| GPT-4o | 675 | 304 | 45 | 197 | 106 | 14 | 9 |
| GPT-4o-mini | 579 | 317 | 22 | 109 | 115 | 15 | 1 |

**Evaluation of OpenAI Codex Agent.** We also evaluated OpenAI Codex, utilizing its agentic scaffolding based on `ChatGPT-5.1`. We conducted a stratified sample of 200 instances, constructed by considering the distribution of refactoring types and executable lines of code (ELOC). Specifically, we selected 100 instances with ELOC $\leq$ 10 and 100 with ELOC > 10. This resulted in 80 *Extract*, 13 *Inline*, 74 *Move*, 26 *Extract+Move*, 4 *Move+Rename*, and 3 *Move+Inline* instances. Codex was provided with full repository access and the same prompts used in our prior evaluation.

As shown in Table 5, Codex successfully completed 151 out of 200 instances (75.5%). It performed well on the three atomic refactoring types, achieving 73 successes out of 80 for *Extract Method*, 12 out of 13 for *Inline Method*, and 53 out of 74 for *Move Method*. Its performance was weaker on compound refactorings, solving only 11 out of 26 *Extract and Move* cases, 2 out of 4 *Move and Rename* cases, and none of the 3 *Move and Inline* cases. Most failures occurred because the model applied a different refactoring than the one requested, such as performing only extraction or only movement, or creating a helper class instead of carrying out the compound refactoring operation.

What's more, **GPT-5.1-Codex** achieves a success rate of 75.5% on our 200-instance sample, which is close to the 74.5% it reports on *SWE-bench-Verified* OpenAI (2025). The similarity between these two results suggests that *SWE-Refactor* poses a comparable level of difficulty, and we believe it is sufficiently challenging for evaluating LLM performance on refactoring tasks.

Table 5: Codex agentic scaffolding results on the 200 samples.

| Model | Total(Success) | EM (80) | IM (13) | MM (74) | EM+MM (26) | MM+RM (4) | MM+IM (3) |
|---|---|---|---|---|---|---|---|
| Codex | 151 | 73 | 12 | 53 | 11 | 2 | 0 |
| GPT-4o | 134 | 59 | 9 | 46 | 17 | 1 | 2 |

## 5 Discussion

**Error Taxonomy.** To analyze failure modes, we sampled 50 refactorings for each of three representative settings: a small code LLM, a general LLM, and a multi-agent workflow. The small code LLM (i.e., CodeLLaMa-7B) failed on nearly all sampled cases, primarily because most outputs ignored the format requirements specified in the prompt, resulting in parsing errors. In contrast, the general LLM (i.e., GPT-4o-mini) was more reliable in following instructions but still showed weaknesses in handling code dependencies and repository-level information. Its major failures included syntax-level errors (e.g., undefined variables and parameter type mismatches) and semantic errors such as moving methods into non-existent files. The multi-agent workflow (using GPT-4o-mini) succeeded in most cases, though its remaining failures often reflected overfitting to the test cases. For example,

generating empty methods that passed compilation and testing but failed AST-Based Refactoring Verification. The observed error patterns highlight the distinct strengths and weaknesses of different LLMs, RAG, and the multi-agent workflow. The results also show that *SWE-Refactor* can assess LLM robustness at multiple levels, from following basic schema in small models to performing repository-level reasoning in multi-agent systems.

**Limitations.** *SWE-Refactor* has three main limitations. First, it focuses only on Java projects. While this limits language diversity, it enables reliable extraction using mature Java-based code analysis tools such as *RefactoringMiner* Tsantalis et al. (2020), RefDiff Silva et al. (2021), and PMD PMD (2025), and provides a valuable complement to existing Python-centric benchmarks. We plan to extend to other languages to support multi-language evaluation. Second, *SWE-Refactor* currently targets method-level refactorings due to their high prevalence in real-world projects Kim et al. (2014); Negara et al. (2013). Higher-level refactorings such as those at the class level are less frequent and often entangled with non-refactoring changes such as bug fixes Penta et al. (2020), which makes extraction more challenging. We aim to include a broader range of refactoring types in the future. Third, although *SWE-Refactor* includes 1,099 pure refactorings from 18 projects, making it one of the largest benchmarks of its kind, the scale is still limited for comprehensive evaluation or fine-tuning of LLMs. We plan to continue expanding the dataset to improve coverage and diversity.

## 6 CONCLUSION

In this work, we present *SWE-Refactor*, a new benchmark specifically designed to evaluate the capabilities of LLMs in code refactoring. *SWE-Refactor* features 1,099 pure, real-world refactorings extracted from 18 diverse Java projects, covering both atomic and compound refactoring types. It ensures high data quality through automated filtering, compilation, and test verification, and includes rich repository-level information to support realistic and comprehensive evaluation. We evaluate 9 widely used LLMs across multiple dimensions, revealing substantial differences in their performance across refactoring types and highlighting the effectiveness of multi-agent prompting strategies. Our results show that large-scale general purpose models like DeepSeek V3 and GPT-4o-mini outperform open-source ones, with DeepSeek V3 achieving the highest success rate. We publicly release all data and results to support future research in LLM-based code refactoring.

## 7 DATA AVAILABILITY

The *SWE-Refactor* data and the code associated with this work can be found in Appendix A.

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

APPENDIX

TABLE OF CONTENTS

## A  DATASET HOSTING

Our *SWE-Refactor* benchmark and experimental results (e.g., code, prompts, and LLM predictions) are available on the following platform:

- **Zenodo**: `https://doi.org/10.5281/zenodo.17196850`

## B  USE OF LARGE LANGUAGE MODELS (LLMS)

Large Language Models (LLMs) were used only to polish the writing. They were not involved in the research design, analysis, or conclusions.

## C  REFACTORING TYPE DEFINITIONS

We define the refactoring types evaluated in this study based on widely accepted descriptions from Fowler's Refactoring Catalog (Fowler, 1999) and *RefactoringMiner* (Tsantalis et al., 2020). These definitions serve as the foundation for identifying and categorizing both basic and compound refactorings in our benchmark.

- **Extract Method.** A code fragment is extracted from an existing method and placed into a newly created method. The original fragment is replaced with a method call. This improves readability, modularity, and reuse, especially when the original method becomes long or performs multiple responsibilities.
- **Move Method.** A method is relocated from one class to another, usually when it relies more on the data of the target class. This improves cohesion and reduces coupling between classes.
- **Inline Method.** A method is removed by replacing its invocations with its body. This is typically done when the method is too simple, no longer adds meaningful abstraction, or is used only once.
- **Extract and Move Method.** A compound refactoring where a code fragment is first extracted into a new method, and the resulting method is then moved to another class (often a superclass). This is useful when the extracted logic is generalizable or better fits in a shared parent class.
- **Move and Rename Method.** A method is moved to a different class and renamed during the process. The renaming helps to align the method name with its new context or to resolve naming conflicts.
- **Move and Inline Method.** A method is first moved to a new class and then inlined at all its call sites. This effectively eliminates the method definition while relocating its logic, typically used when the method becomes redundant after reorganization.

Table 6: Overview of Java projects used in the construction of *SWE-Refactor*.

| Project | # Stars | # Commits | # Pure Refactorings |
|---------|--------|-----------|---------------------|
| checkstyle | 8,462 | 14,606 | 91 |
| pmd | 4,988 | 29,117 | 125 |
| commons-lang | 2,776 | 8,404 | 59 |
| hibernate-search | 512 | 15,716 | 89 |
| junit4 | 8,529 | 2,513 | 18 |
| commons-io | 1,020 | 5,455 | 93 |
| javaparser | 5,682 | 9,607 | 56 |
| junit5 | 6,523 | 8,990 | 105 |
| hibernate-orm | 6,091 | 20,638 | 63 |
| mockito | 15,032 | 6,236 | 4 |
| gson | 24080 | 2135 | 21 |
| guava | 51140 | 7068 | 300 |
| jadx | 45589 | 2512 | 18 |
| zxing | 33605 | 3832 | 21 |
| shiro | 4402 | 4222 | 2 |
| shenyu | 8663 | 3680 | 22 |
| shardingsphere-elasticjob | 8211 | 2473 | 3 |
| hertzbeat | 6665 | 2632 | 9 |
| **Total** | **241,970** | **149,836** | **1099** |

- **Extract Variable.** Extracts part of an expression or a literal value into a new local variable. This improves readability and allows reuse of the extracted value. It is often applied to clarify complex expressions or remove duplication.

- **Rename Method.** Changes the name of a method to better reflect its purpose or conform to naming conventions. This improves code readability and maintainability. All call sites must be updated accordingly.

- **Move Class.** Relocates a class from one package or module to another. This helps improve package organization and reduce module dependencies. All references and imports must be updated.

- **Rename Class.** Changes the name of a class to better reflect its role or to align with naming standards. This refactoring improves clarity and consistency. The renaming may also require updating file names and documentation.

## D  PROJECT SELECTION AND REFACTORING DISTRIBUTION

We selected 18 Java projects previously used in change history tracking studies (Grund et al., 2021; Jodavi & Tsantalis, 2022; Hasan et al., 2024) based on three key criteria. First, the projects span diverse application domains, offering broad coverage of real-world software development practices. Second, each project has a rich development history, with over 2,000 commits, increasing the likelihood of discovering meaningful refactoring activities. Third, we ensured that the selected projects could be compiled and tested successfully after manual resolution of build issues, making it feasible to verify the correctness of the generated refactorings.

Table 6 presents the selected Java projects along with the number of extracted pure refactorings for each project.

# E PROMPT TEMPLATES FOR DIFFERENT REFACTORING TYPES

- **Prompt Template for Extract Method, Inline Method Refactoring.**

```
Task:
You are an expert software engineer. You are given a code to be
refactored. The objective is to refactor this code by performing
given refactoring operation. This refactoring will improve code
readability, maintainability, and modularity.
Code to be Refactored:
{code_to_refactor}
Class content:
{class_content}
Refactoring Operation:
{refactoring_operation}
Call Relationship:
{call_relationship}
Instructions:
1. Analyze the provided code and class content, apply relevant
   refactoring operation to the code to be refactored.
2. If refactoring is performed, output the refactored_method_code
   in the following format:
#########################
refactored_method_code
#########################
```

- **Prompt Template for Move Method, Move And Rename Method Refactoring.**

```
Task:
You are an expert software engineer. You are given a code to be
refactored. The objective is to refactor this code by performing
given refactoring operation. This refactoring will improve code
readability, maintainability, and modularity.
Code to be Refactored:
{code_to_refactor}
Class content:
{class_content}
Refactoring Operation:
{refactoring_operation}
Call Relationship:
{call_relationship}
Project Structure:
{project_structure}
Instructions:
1. Analyze the provided code, class content, and project
structure, apply move method refactoring to the code to be
refactored, output the target file path, moved class code,
and refactored method code. Need to move to an existing
java file
The moved method code should be updated to the public
static method. The refactored method code should use the
moved class to call the moved method.
The target file path should be the path of the existing class
where the method is moved to.
2. If refactoring is performed, output the target file path,
moved class code, and refactored method code in the following
format:
#########################
target_file_path
#########################
moved_class_code
#########################
refactored_method_code
#########################
```

- **Prompt Template for Move And Inline Method Refactoring.**

```
Task:
You are an expert software engineer. You are given a code to be
refactored. The objective is to refactor this code by performing
given refactoring operation. This refactoring will improve code
readability, maintainability, and modularity.
Code to be Refactored: {code_to_refactor}
Class content: {class_content}
Refactoring Operation: {refactoring_operation}
Call Relationship: {call_relationship}
Project Structure: {project_structure}
Instructions:
1. Analyze the provided code, class content, and project
structure, apply relevant refactoring operation to the
code to be refactored, output the target file path.
2. If refactoring is performed, output the refactored class code
in the following format:
#########################
target_file_path
#########################
refactored_class_code
#########################
```

- **Prompt Template for Extract And Move Method Refactoring.**

```
Task:
You are an expert software engineer. You are given a code to
be refactored. The objective is to refactor this code by
performing given refactoring operation. This refactoring will
improve code readability, maintainability, and modularity.
Code to be Refactored: {code_to_refactor}
Class content: {class_content}
Refactoring Operation: {refactoring_operation}
Call Relationship: {call_relationship}
Project Structure: {project_structure}
File Path Before Refactoring:
{file_path_before_refactoring}
Instructions:
1. Analyze the provided code, class content, and project
structure, apply relevant refactoring operation to the code
to be refactored, and you need move the
extracted method to another existing java file, output the
target file path, extracted method code, refactored method code
after refactoring.
The extracted method code should be the public static method.
The refactored method code should use the moved class to call the
extracted method.
The target file path should be the path of the existing class
where the method is moved to.
2. If refactoring is performed, output the refactored class code
in the following format:
#########################
target_file_path
#########################
extracted_method_code
#########################
refactored_method_code
#########################
```

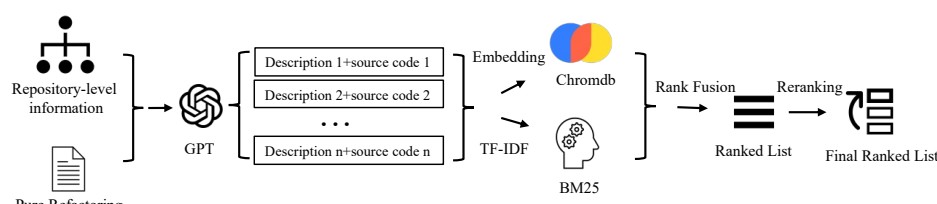

Figure 4: RAG Construction and Retrieval Pipeline.

## F    RAG CONSTRUCTION FOR REFACTORING RETRIEVAL

To support more accurate LLM-based code refactoring, we design a retrieval-augmented generation (RAG) pipeline. As shown in Figure 4, it consists of four main steps: preparing the inputs, generating descriptions, retrieving relevant examples using both text and embedding similarity, and merging the results to find the most suitable matches.

### STEP 1: PREPARING INPUTS FROM REFACTORING COMMITS

We apply our pipeline (Section 3.3) to the *Refactoring Oracle Dataset* (Tsantalis et al., 2020), which contains over 12,000 refactorings collected from 547 commits across 188 open-source Java projects. This dataset has been widely used to evaluate refactoring detection tools and covers diverse projects and refactoring types. Using our pipeline, we extract a set of 905 pure method-level refactorings from this dataset. To save time, we do not perform compilation or test verification on these examples, as they are intended to illustrate refactoring strategies for retrieval rather than for correctness evaluation.

For each refactoring, we also collect repository-level information such as the file path, class definition, method signature, and the method's direct callers and callees. These elements form the foundation of our retrieval database.

### STEP 2: GENERATING DESCRIPTIONS OF REFACTORING EXAMPLES

For each example, we use `gpt-4o-mini-0125` to generate a short natural language description that summarizes the method's functionality and surrounding structural information. The model takes as input the method before refactoring, its enclosing class, and the bodies of its direct callers and callees. These descriptions help guide retrieval by expressing the purpose and behavior of the method in a form that complements its code.

We use the following prompt template:

```
{Method Code}
{Caller/Callee Code}
{Class Code}
Please give a short, succinct description to situate this
code within the class.
```

Here, {`Method Code`} is the code to be refactored, {`Caller/Callee Code`} includes the full bodies of its direct callers and callees, and {`Class Code`} provides the signature and body of the class containing the method.

### STEP 3: CONSTRUCTING A SEARCHABLE DATABASE OF REFACTORING EXAMPLES

To support downstream retrieval, we construct a database of refactoring examples, where each entry includes both the code and its generated description. We index the database using two complementary methods to support both lexical and semantic similarity.

For text-based indexing, we apply BM25 (Robertson et al., 2009), which ranks examples based on token overlap and structural similarity in the combined code and description.

For semantic indexing, we use `all-MiniLM-L6-v2` (Reimers & Gurevych, 2019) to generate vector embeddings for each example. This enables similarity computation based on meaning, not just syntax.

### STEP 4: MERGING AND RERANKING THE RESULTS

When a new refactoring task is issued, both text-based and embedding-based retrieval models produce independent similarity-ranked lists based on the input query. To combine these results, we apply the Reciprocal Rank Fusion (RRF) algorithm (Cormack et al., 2009), which merges the rankings by assigning higher scores to examples that appear near the top of either list.

To further improve ranking quality, we apply a reranking step that refines the similarity assessment between the query and the retrieved examples. This step helps prioritize examples that are both lexically and semantically aligned with the input.

Finally, we select the top 3 ranked examples to serve as few-shot prompts, guiding the LLM to generate accurate and structurally relevant refactored code.

## G   WORKFLOW FOR MULTI-AGENT

To examine how multi-agent LLM workflows perform in automated code refactoring, we design a flexible agent-based system and evaluate it using our benchmark, *SWE-Refactor*. The workflow is composed of two core agents: a *Developer Agent* and a *Reviewer Agent*. These agents communicate and collaborate through iterative reasoning and feedback.

### DEVELOPER AGENT: GENERATION AND REFINEMENT

The *Developer Agent* is tasked with analyzing source code and generating refactored code. It has three main capabilities: *Analyzing*, *Programming*, and *Enhancing*. To support these tasks, the agent can invoke a variety of utility methods, such as retrieving project structure, reading source files, obtaining class body, or getting callers and callees. These methods are implemented through command-line tools or APIs from static analysis frameworks. After collecting the necessary information, the agent composes a prompt combining structural analysis and submits it to the LLMs to produce a refactored version of the target method. The agent can also iteratively improve its output by incorporating feedback received from the *Reviewer Agent*.

### REVIEWER AGENT: EVALUATION AND FEEDBACK

The *Reviewer Agent* is responsible for assessing the quality of the generated refactoring. It performs this assessment by applying static analysis tools, including a refactoring detector (e.g., *Refactoring-Miner* (Tsantalis et al., 2020)) and a style checker (e.g., Checkstyle (Checkstyle Team, 2024)) to detect code smells or violations of coding conventions. Based on this analysis, the *Reviewer Agent* generates feedback indicating whether the refactoring is valid, and if not, what aspects should be improved. This feedback is then sent back to the *Developer Agent* for further refinement.

