# OpenReview forum: "SWE-Refactor: A Repository-Aware Benchmark for Evaluating LLMs on Real-World Code Refactoring"
_ICLR.cc/2026/Conference — Submitted to ICLR 2026_

### Official Review · Reviewer_5ytC · 2025-10-20

**Soundness:** 3
**Presentation:** 4
**Contribution:** 2
**Rating:** 2
**Confidence:** 4

**Summary:**

This paper introduces SWE-Refactor, a benchmark for evaluating LLMs on code refactoring using 1099 "pure" instances from 18 real-world Java projects. The contributions are ensuring data purity by filtering out non-refactoring changes and rigorously verifying each instance for correctness using the projects' full test suites. The benchmark provides repository-level context and covers both atomic and compound refactoring types. An evaluation of 9 LLMs finds that a multi-agent workflow performs best, but with a modest top success rate of 52.7%.

**Strengths:**

- Originality: The paper addresses a clear gap in existing work. Current refactoring benchmarks are often noisy, lack repository-level context, and are heavily biased toward Python. SWE-Refactor’s focus on pure, verifiable, context-rich refactorings in Java is an original contribution.
- Quality: Verifying each refactoring via compilation and the full test suite is a significant improvement over prior benchmarks, ensuring models are evaluated on the refactoring task itself.
- Clarity: The paper is well-written, and the problem, solution, and construction pipeline are all clearly articulated.

**Weaknesses:**

- Limited Scope: The benchmark's primary weakness is its focus on six well-defined, syntactic refactoring types (e.g., Extract Method) that are already reliably automated by modern IDEs. It misses the opportunity to evaluate models on more complex, semantic refactors that lack tool support.
- Outdated Model Evaluation: For a paper targeting a 2026 conference, the model selection is not representative of the current state of the art. While the evaluation includes 2024 models like GPT-4o-mini and DeepSeek-V3, the rapid advances throughout 2025 have introduced significantly more powerful models (e.g. Claude 4 series, or GPT reasoning models).
- Lack of Difficulty: Related to the point above, the benchmark's difficulty appears mismatched with current state-of-the-art agentic frameworks. While the paper's top score is 52.7% with a multi-agent workflow, other recent work has shown much higher performance on similar tasks. For instance, the MANTRA multi-agent framework, which is designed for this type of method-level refactoring, reports an 82.8% success rate on a similar "pure refactoring" dataset [1]. And the public leaderboard for the popular aider coding tool shows a recent version of Claude 3.5 Sonnet achieving 92.1% accuracy on its refactoring benchmark [2]. This suggests the benchmark may already be largely solved by SOTA agents, limiting its long-term utility for a 2026 conference.
- Contradictory Scalability Claims: The paper claims its "fully automated... pipeline" is an advantage. However, this pipeline produced only 1099 examples. The authors themselves concede this scale is "still limited". This small size fails to convincingly demonstrate the claimed scalability.

---
[1] Xu, Y., Lin, F., Yang, J., Chen, T. H., & Tsantalis, N. (2025). MANTRA: Enhancing Automated Method-Level Refactoring with Contextual RAG and Multi-Agent LLM Collaboration. arXiv preprint arXiv:2503.14340.

[2] Aider. (2025). Refactoring Leaderboard. https://aider.chat/docs/leaderboards/refactor.html

**Questions:**

1. Could the authors provide results from more recent SOTA models (e.g., GPT-4o, Claude 4.x Sonnet, Gemini 2.5 Pro) and agentic scaffolding (e.g., OpenAI Codex, Claude Code)? This is crucial to demonstrate that the benchmark is not already "solved" and remains a useful challenge.
2. What was the main bottleneck in the "automated pipeline" that limited the dataset to 1099 instances? Specifically, what was the filtering rate? (i.e., how many candidate refactorings were discarded by PurityChecker for being "impure"?) This would clarify the trade-off being made between purity and scale.
3. Given that the included refactoring types are largely automated by IDEs, can the authors elaborate on the practical value of having LLMs solve these specific tasks, as opposed to focusing on more complex, semantic refactors that lack any tool support?

---

> ### Author Response · Authors · 2025-11-20
> **Authors' Rebuttal 1**
>
> >**Weakness 2 & Weakness 3 & Question 1. Could the authors provide results from more recent SOTA models (e.g., GPT-4o, Claude 4.x Sonnet, Gemini 2.5 Pro) and agentic scaffolding (e.g., OpenAI Codex, Claude Code)? This is crucial to demonstrate that the benchmark is not already "solved" and remains a useful challenge.**
>
> Thank you for the insightful comment. Following the suggestion, we conducted additional experiments using (1) the SOTA model **GPT-4o**, and (2) an **agentic scaffolding setup using OpenAI Codex** (`gpt-5.1-codex`, agent mode, high-reasoning). **We have also updated the paper to include these new results**. Below, we summarize the extended evaluation results.
>
> ---
>
> ### **1. Evaluating SOTA model: GPT-4o**
>
> We extended our main experiment by replacing the base model with GPT-4o for the agent approach. Table 1 summarizes the full benchmark results.
>
> - GPT-4o achieves **675 out of 1,099 successful refactorings (61.4%)**, an improvement over GPT-4o-mini, which achieves **579 out of 1,099 (52.7%)**.
> - The largest gains appear in navigation-intensive refactoring types, such as **Move Method** and **Move and Inline Method**, suggesting that GPT-4o’s stronger reasoning and repository-navigation capabilities help the agent locate the correct files and apply the required edits more reliably.
> - The complete GPT-4o result files have been uploaded to Zenodo:
>   https://zenodo.org/records/17196850
>
> ---
>
> ### **Table 1. GPT-4o Results on Full SWE-Refactor (1,099 instances)**
>
> | Model              | Total (Success) | Extract Method | Inline Method | Move Method | Extract and Move Method | Move and Rename Method | Move and Inline Method |
> |--------------------|----------------|----------------|---------------|-------------|--------------------------|------------------------|------------------------|
> | **gpt-4o result**       | 675            | 304            | 45            | 197         | 106                      | 14                     | 9                      |
> | **gpt-4o-mini result**  | 579            | 317            | 22            | 109         | 115                      | 15                     | 1                      |
>
> **2. Evaluating agentic scaffolding with OpenAI Codex**
>
> We chose **OpenAI Codex** as it is one of the latest agentic scaffolds and is based on the newest ChatGPT 5.1. Given the limited time, we manually ran a stratified sample of **200 instances** for the Codex evaluation. We constructed the sample by considering the distribution of refactoring types and executable lines of code (ELOC) in SWE-Refactor. The benchmark has an average ELOC of 17.55 and a median of 10, so we selected **100 instances with ELOC ≤ 10** and another **100 instances with ELOC > 10**. As a result, we selected **80 Extract Method**, **13 Inline Method**, **74 Move Method**, **26 Extract and Move**, **4 Move and Rename**, and **3 Move and Inline** instances.
>
> We provided Codex with a complete prompt that included the method code before refactoring, the file path, and the refactoring operation that needed to be performed (the same prompt used in our prior evaluation). We gave Codex full access to the repository.
>
> Of the **200 instances**, Codex successfully completed **151 instances (75.5%)**. The detailed results of OpenAI Codex and the corresponding output files have been uploaded to Zenodo:
> https://zenodo.org/records/17196850
>
> OpenAI Codex performed well on the three atomic refactoring types, achieving **73/80 Extract Method**, **12/13 Inline Method**, and **53/74 Move Method**. Its performance was weaker on compound refactorings, solving only **11/26 Extract and Move**, **2/4 Move and Rename**, and **0/3 Move and Inline** cases. Most failures occurred because the model applied a different refactoring than the requested one, such as performing only extraction or only movement, or creating a helper class instead of carrying out the compound refactoring operation.
>
> What’s more, GPT-5.1-Codex achieves a success rate of **75.5%** on our 200-instance sample, which is close to the **74.5%** it reports on SWE-bench-Verified [1]. **The similarity between these two results suggests that SWE-Refactor poses a comparable level of difficulty, and we believe it is sufficiently challenging for evaluating LLM performance on refactoring tasks.**
>
> ---
>
> ### **Table 2. Codex Agentic Scaffolding Results on the 200 Samples**
>
> | Model | Total Success | Extract Method (80) | Inline Method (13) | Move Method (74) | Extract and Move Method (26) | Move and Rename Method (4) | Move and Inline Method (3) |
> |--------|--------------|--------------------|--------------------|------------------|------------------------------|----------------------------|----------------------------|
> | **Codex** | 151 | 73 | 12 | 53 | 11 | 2 | 0 |
> | **GPT-4o** | 134 | 59 | 9 | 46 | 17 | 1 | 2 |
>
> [1] Introducing upgrades to Codex, https://openai.com/index/introducing-upgrades-to-codex/

---

> ### Author Response · Authors · 2025-11-20
> **Authors' Rebuttal 2**
>
> >**Weakness 4 & Question 2. What was the main bottleneck in the "automated pipeline" that limited the dataset to 1099 instances? Specifically, what was the filtering rate? (i.e., how many candidate refactorings were discarded by PurityChecker for being "impure"?) This would clarify the trade-off being made between purity and scale.**
>
> Thanks for your valuable comment. The main bottleneck in our automated pipeline is **not the PurityChecker**. The real bottlenecks come from the later steps in **ensuring code quality**: checking whether the target file exists, whether the project can compile and pass tests, and whether the refactoring is covered by tests.
>
> We first used RefactoringMiner to **collect 29,754 refactorings**, including both pure and impure refactorings. We then apply the PurityChecker to filter out impure refactorings, which leaves **16,482 pure candidates**. In the next step, we verify the existence of the target files for the Move Method, Extract and Move, Move and Inline, and Move and Rename refactorings. This step is important because these refactorings may move a method into a newly created class, and it is difficult for an LLM or an agent to predict the name or location of such new classes (for automated evaluation). After applying this filter, **7,511 instances remain**.
>
> We then check whether the commit of each refactoring can compile and pass the tests, which further reduces the set to **3,711 instances**. The final step ensures that each refactoring is covered by the test cases. After applying this last filter, **we obtain 1,099 high-quality pure refactorings**.
>
> >**Weakness 1 & Question 3. Given that the included refactoring types are largely automated by IDEs, can the authors elaborate on the practical value of having LLMs solve these specific tasks, as opposed to focusing on more complex, semantic refactors that lack any tool support?**
>
> Thank you very much for this valuable comment.
>
> We would first like to clarify that **current IDEs cannot fully automate even common refactorings such as Extract Method or Move Method**. In practice, developers must still make several non-trivial decisions: selecting the exact code block to extract, choosing meaningful method names, determining appropriate parameterization, and, for Move Method, identifying the correct destination class. **Modern IDEs only execute the final mechanical transformation, and they do not perform any of these decisions automatically**.
>
> In contrast, **SWE-Refactor evaluates the entire refactoring workflow**. Given a method and a target refactoring activity, the model must: (1) identify the correct refactoring opportunity, (2) reason about where the transformation should occur, and (3) apply the behavior-preserving change correctly. These reasoning steps are precisely the components that are still fully manual for developers today.
>
> Second, **SWE-Refactor includes more complex compound refactorings**, such as Extract and Move Method, and Move and Inline Method. These cases require coordinated multi-step transformations and provide a stronger test of the model's capabilities. Our Codex results show that current agentic models still struggle with these compound refactorings.

---

> > ### Author Response · Authors · 2025-11-27
> >
> > Dear reviewer **5ytC**,
> >
> > We would like to further clarify and demonstrate how such refactorings are typically carried out in **IDE tools** and in our **multi-agent evaluation setting**. We provide an example below.
> >
> > Take the *Extract And Move Method* as an example. A traditional refactoring tool (e.g., **JetBrain IDE**) requires the developer to manually specify several decisions:
> >
> > 1. **which exact code block should be extracted**,
> > 2. **what the new method should be named**,
> > 3. **which class the extracted method should be moved to**, and
> > after these steps, the developer must still
> > 4. **rebuild the project**,
> > 5. **run tests**, and
> > 6. **fix any issues introduced during the process**.
> >
> > With our benchmark, **we can evaluate multi agent techniques that require two inputs**:
> > 1. **the original method** and
> > 2. **the refactoring type “Extract And Move Method.”**
> >
> > **The agent automates the entire process and runs compilation and tests.**
> >
> > We hope that these updates adequately address your concerns and provide a clearer picture of the difficulty and practical value of our benchmark. In light of these changes, we kindly request that you reconsider your overall assessment.
> >
> > Thanks for your time and consideration.

---

### Official Review · Reviewer_aF72 · 2025-11-01

**Soundness:** 3
**Presentation:** 4
**Contribution:** 4
**Rating:** 6
**Confidence:** 4

**Summary:**

In this work, the authors propose a benchmark for code refactoring. Given that this task is heavily reliant on memory (i.e., structure and class hierarchies in a code repository), as well as inter-component dependencies; it captures a whole different nuance in coding agents (as opposed to bug-fixing as in SWE-Bench et al.). The authors present a methodical approach for data collection, directly addressing the limitations of existing code refactoring benchmarks; while designing a two-agent, iterative feedback loop to incorporate repository awareness. However, the evaluation remains flawed (discussed more in Weaknesses), which fails to capture the true complexity of refactoring quality and behavioral preservation.

**Strengths:**

1. Code refactoring is a complex SE task that demands systemic understanding of class hierarchies and inter-file dependencies. Through this work, the authors correctly shift coding agent evaluation away towards context-dependent reasoning, which is essential for agent development.

2. The data collection is methodical and significantly improves over the existing refactoring benchmarks, which typically suffer from ambiguous or noisy data.

**Weaknesses:**

1. The Reviewer Agent appears to rely on: static analysis, which cannot verify functional correctness; and pre-existing test suite, which inherently assumes completeness of the test suite. The other metric, CodeBLEU, is in itself brittle.

2. The multi-agent improvements are notable, but the discussion does not analyze why they succeed (e.g., self-critique vs. feedback loops) or whether such workflows generalize across domains. A breakdown of iteration counts or failure recoveries would add clarity.

3. While the methodical data collection is a strength, the benchmark itself is limited to Java; and would greatly benefit from being designed for multiple programming languages.

**Questions:**

1. Beyond eliminating noisy refactoring commits, how did the authors ensure that the original code's test suite, used for validation, is stable and functionally complete?

2. Besides the final correctness and CodeBLEU score, did the authors quantify the efficiency of the iterative process? Specifically, what are the metrics for the token consumption per task and the average number of iterations required to reach a correct solution?

3. What are the technical barriers and necessary tool replacements for adapting SWE-Refactor to a second language?

---

> ### Author Response · Authors · 2025-11-20
> **Authors' Rebuttal 1**
>
> >**Weakness 1 & Question 1. Beyond eliminating noisy refactoring commits, how did the authors ensure that the original code's test suite, used for validation, is stable and functionally complete?**
>
> Thank you for the question, and we are sorry for any potential confusion. We did not manually/automatically remove any code from the collected data. When we identify that a commit contains noisy code changes (i.e., not pure refactoring changes), we skip the commit. We only collect the commits that contain only pure refactoring changes.
>
> To validate these commits, we then perform two checks. First, we verify that the entire repository can be built and that all tests pass, which ensures basic stability. Second, we check that the refactored method is covered by the existing tests so that its behavior is exercised.
>
> Importantly, our pipeline does **not** rely solely on the completeness of the test suite. All collected refactorings must also pass strict AST-based purity checks (PurityChecker) and AST-based refactoring verification (RefactoringMiner). These structural and behavioral validations ensure that the included refactorings are behavior-preserving even if the project’s test suite may not be fully complete.
>
> >**Weakness 1 & Question 2. Besides the final correctness and CodeBLEU score, did the authors quantify the efficiency of the iterative process? Specifically, what are the metrics for the token consumption per task and the average number of iterations required to reach a correct solution?**
>
> Thank you for the question. In this work, our focus is on assessing the final correctness and the quality of the generated code, so we did not include dedicated metrics for the efficiency of the iterative process. We did not track token consumption per task or the average number of iterations required to reach a correct solution. These measurements would require instrumenting the system to log token-level usage and iteration counts across all runs, which is not currently part of our experimental setup. We agree that such information would be valuable for understanding the cost and efficiency of iterative repair, and we plan to include these measurements in future work.
>
> To get some insights, we conducted a small preliminary measurement of the agent approach on the Checkstyle project to examine token consumption and iteration counts. In this project, the agent needs an average of **9.35 iterations**, where each iteration is a simple query–response step, such as calling a tool (e.g., get_project_structure) and reading the returned output. Overall, the process consumes approximately **151K tokens per instance**, which corresponds to an estimated **$0.03 using gpt-4o-mini**.
>
> > **Weakness 2: The multi-agent improvements are notable, but the discussion does not analyze why they succeed (e.g., self-critique vs. feedback loops) or whether such workflows generalize across domains. A breakdown of iteration counts or failure recoveries would add clarity.**
>
> Thank you for the comment. In the paper, we already included an analysis of 50 representative refactorings to understand the common mistakes made by the multi-agent workflow, and we further manually examined these cases to understand why the multi-agent workflow performs well.
> The main reason is the feedback loops between the Reviewer Agent and the Developer Agent, where static analysis tools and execution results provide concrete guidance, and the Developer Agent adjusts the generated refactoring accordingly. In these 50 instances, the multi-agent workflow completed 32 refactorings successfully. Among them, 15 were completed immediately without any review, and the remaining 17 were completed through several rounds of review and correction. We will clarify this process in the paper.

---

> ### Author Response · Authors · 2025-11-20
> **Authors' Rebuttal 2**
>
> >**Weakness 3 & Question 3. What are the technical barriers and necessary tool replacements for adapting SWE-Refactor to a second language?**
>
> Thank you for the question. Our pipeline consists of five main components:
>
> 1. **RefactoringMiner** for detecting refactorings (including both pure and impure cases),
> 2. **PurityChecker** for filtering pure refactorings,
> 3. **Eclipse JDT** for static code analysis (such as call graph extraction),
> 4. **compilation and test execution** for validating project buildability, and
> 5. **JaCoCo** for collecting coverage information.
>
> These components together produce high-quality refactoring instances.
>
> To adapt SWE-Refactor to a second programming language, several components in this pipeline would need to be replaced. The first and largest component is the refactoring detector. **RefactoringMiner is specific to Java**, so an equivalent detector would be needed. For Python, **PYREF** [1] is available, and for C++, **RefactoringMiner++** [2] exists. However, both tools currently support fewer refactoring types and have lower precision (e.g., PYREF supports 9 refactoring types with 89.6% precision).
>
> The second major component requiring replacement is **PurityChecker**. To the best of our knowledge, there is no existing tool for checking the purity of refactorings in other languages. Extending the formal rules used in the Java version to other languages is possible, but would require engineering effort.
>
> Other components, such as static analysis frameworks and coverage tools, are easier to replace because most languages have mature ecosystems that provide these capabilities.
>
> We want to note that **more than 95% of existing SE benchmarks are Python-centric** (e.g., HumanEval, MBPP, SWE-Bench, RefactorBench). **SWE-Refactor fills an important gap** by providing a Java benchmark that reflects the languages used in many enterprise software systems.
>
> Although SWE-Refactor is constructed for Java, it also provides a controlled environment for developing LLM-based refactoring detectors and purity checkers. These components—such as identifying refactoring opportunities, reasoning about structural changes, and distinguishing pure from impure edits—are *largely language-agnostic reasoning tasks*. By training and evaluating LLM-based capabilities on SWE-Refactor, researchers can build solutions that are potentially transferable to other languages when paired with appropriate language-specific parsers.
>
> In this sense, SWE-Refactor serves not only as a Java benchmark but also as a testbed for developing the next generation of LLM-driven refactoring tools (for both detecting and performing refactoring) across multiple programming languages.
>
> [1] Atwi, Hassan, et al. "Pyref: Refactoring detection in python projects." 2021 IEEE 21st International Working Conference on Source Code Analysis and Manipulation (SCAM). IEEE, 2021.
> [2] Ritz, Benjamin, Aleksandar Karakaš, and Denis Helic. "Refactoring Detection in C++ Programs with RefactoringMiner++." Proceedings of the 33rd ACM International Conference on the Foundations of Software Engineering. 2025.

---

### Official Review · Reviewer_GqSu · 2025-11-01

**Soundness:** 3
**Presentation:** 3
**Contribution:** 3
**Rating:** 6
**Confidence:** 4

**Summary:**

This paper introduces SWE-Refactor, a new benchmark for evaluating LLMs on real-world Java code refactoring tasks. The benchmark contains 1,099 pure refactorings from 18 Java projects, covering both atomic and compound refactoring types. This paper propose fully automated pipeline for benchmark construction and evaluate 9 popular LLMs, finding that DeepSeek-V3 achieves best performance with 41.58% success rate.

**Strengths:**

- This paper is well-written and addresses important limitations such as supporting compound refactorings, ensuring pure refactorings without noise, and providing an automated construction pipeline.
- This method has built comprehensive evaluation metrics and conducted extensive experiments, which provide the community with valuable insights.

**Weaknesses:**

- This benchmark focuses only on Java limits generalizability. While authors justify this choice, it's significant limitation for comprehensive LLM evaluation.
- This benchmark contains only 1,099 samples across 6 refactoring types, some categories have very few examples, which may raise some biases.

**Questions:**

- How do you ensure RefactoringMiner's 99% precision  translates to correct ground truth, given potential tool errors?
- Could you provide quantitative comparison of data quality between SWE-Refactor and RefactorBench on overlapping refactoring types?

---

> ### Author Response · Authors · 2025-11-20
> **Authors' Rebuttal 1**
>
> > **Weakness 1: This benchmark focuses only on Java limits on generalizability. While authors justify this choice, it's significant limitation for comprehensive LLM evaluation.**
>
> Thank you for the comment. We agree that focusing only on Java limits language generalizability. As stated in the paper, this choice enables reliable extraction using mature Java-based code analysis tools such as RefactoringMiner [1], RefDiff [2], and PMD [3], which is important for ensuring high-quality and trustworthy ground truth. At the same time, existing software engineering benchmarks already suffer from a significant imbalance in programming languages, and prior work [4] shows that 95.6% of recent benchmarks are built exclusively on Python. By introducing a benchmark on Java, SWE-Refactor broadens evaluation beyond the current Python-centric landscape and better reflects languages used in large-scale enterprise and open-source systems. While the current version focuses on Java, the methodology itself is not tied to any specific language, and we plan to extend it to additional languages once similarly mature refactoring-detection tools become available.
>
> > **Weakness 2: This benchmark contains only 1,099 samples across 6 refactoring types, some categories have very few examples, which may raise some biases.**
>
> Thank you for the comment. The distribution of the data is not a limitation of our construction pipeline, but a reflection of real refactoring practice. In real projects, compound refactorings (e.g., Extract And Move Method) occur less frequently than atomic ones, and SWE-Refactor preserves this natural frequency rather than synthesizing or oversampling rare cases.
> In our dataset, the three compound categories contain a total of 177 instances, accounting for 16.1% of all 1,099 refactorings, which accurately reflects their real-world frequency. SWE-Refactor contains 1,099 real refactoring instances extracted from large open-source projects, making it **one of the largest refactoring-focused datasets**. Existing benchmarks in this space are substantially smaller, such as RefactorBench [5] with 100 instances and ref-Dataset [6] with 180 examples (for all refactoring types combined). By comparison, SWE-Refactor provides both higher scale and broader coverage across six refactoring types.
>
>
> > **Question 1: How do you ensure RefactoringMiner's 99% precision translates to correct ground truth, given potential tool errors?**
>
> Thanks for your insightful comment. While the result of RefactoringMiner may not be perfect, we apply multiple verification stages to ensure the correctness of the ground truth. RefactoringMiner is the first step in our extraction process and it alone does not determine the final ground truth labels. Our verification includes multiple steps. After ReactoringMiner, we apply PurityChecker to revalidate the detected refactoring and apply a stricter filter that ensures the commit contains only the refactoring and no other behavioral changes. Finally, we ensure that the project can compile and pass developer written tests, and whether the refactoring is covered by the tests. Together, these verification steps help ensure the quality of the collected data.
>
> Before batch running the data collection process on all the projects, we also performed a manual spot check on the first project (Checkstyle) that we collected to confirm that the detected refactorings match the actual code changes. The manual inspection did not reveal any consistent mismatches, which gave us additional confidence in the correctness of our automated verification-and-filtering pipeline.

---

> ### Author Response · Authors · 2025-11-20
> **Authors' Rebuttal 2**
>
> >**Question 2: Could you provide a quantitative comparison of data quality between SWE-Refactor and RefactorBench on overlapping refactoring types?**
>
> Thanks for your comment. Below, we first summarize the prior work, RefactorBench, and then present a **quantitative comparison** of the overlapping refactoring types with our SWE-Refactor.
>
> In short, upon our manual inspection, we find that RefactorBench has **only one overlapping refactoring type** (i.e., *Move Method*) with our SWE-Refactor, and RefactorBench’s tests **only check code structure** (e.g., whether the code contains an import statement like `assertTrue(code, "import re")`). They do not validate the refactoring's **functional behavior and correctness**.
>
> ### **Overview of RefactorBench**
>
> RefactorBench contains 100 instances, each consisting of an instruction, related test cases, and a raw repository. However, upon *manual inspection*, we find that the provided test cases **only verify the structural properties** of the generated code.
>
> For example, the tests check only whether a file contains a specific import statement (e.g., `"import re"`) or whether a target class includes a method with a particular name. These tests do not validate whether a refactoring actually occurred, nor its **functional correctness**.
>
> One test case from RefactorBench for the new-inventory-patterns task [7] is shown below.
> ```python
> def test_import_re_exists(self):
>     file_path = '../lib/ansible/inventory/patterns.py'
>     self.assertTrue(os.path.exists(file_path), f"{file_path} does not exist")
>     …
>     self.assertTrue(import_found, "Import 'import re' not found in patterns.py")
> ```
>
> ### **Refactoring Types in RefactorBench**
>
> Since RefactorBench does not annotate refactoring types, we manually classified all 100 instances into 7 atomic refactoring types and one non-refactoring category. Table 1 reports the distribution.
>
> A notable observation is that **9 out of 100 instances are not refactorings but feature additions.** For example:
>
> ```text
> "Add the none handling based on django/db/models/fields/__init__ to the duration string function in utils/duration so we can just call the function there. add a test_none in the appropriate place for the none handling too."
>
> "add a new PathTraversal exception that extends the suspicious operation exception and replaces some of the SuspiciousFileOperation calls. Update the repository accordingly"
> ```
>
>
> **Table 1. Distribution of refactoring instances and types in the prior study, RefactorBench.**
> | Refactoring Type | Move Method | Add Parameters | Move Class | Rename Class | Rename Method | Spilt Method | Merge Method | Not Refactoring |
> |------------------|-------------|----------------|------------|--------------|---------------|--------------|--------------|-----------------|
> | **# instances**   | 26          | 21             | 8          | 8            | 23            | 1            | 4            | 9               |
>
> ### **Overlapping Refactoring Types**
> As shown in Table 1, across all 7 refactoring types in RefactorBench, only **Move Method** overlaps with SWE-Refactor. Therefore, we further designed four metrics to quantitatively compare the data quality of the **Move Method** between RefactorBench and our SWE-Refactor:
>
> - **Specified Move Target**: Whether the instruction already specifies the exact target location for the method move (i.e., whether the model still needs to identify the refactoring opportunity).
> - **Existing Target**: Whether the target classes actually exist in the repository, rather than requiring the model to create new files.
> - **Functional Tests**: Whether functional test cases are provided to validate the behavioral correctness of the refactoring.
> - **Executability**: Whether the repository is executable, meaning it can be built and run without fixing further issues.
>
> We use these four metrics because **Move Method** is a repository-grounded refactoring task in which the main challenge is selecting a valid target class in the existing codebase, not simply applying a transformation. **Specified Move Target** checks whether the model needs to find the move location itself or just follow a given instruction. **Target Class Existence** ensures that the target location is a valid Java class in the repository (as reflected in the developer’s ground truth). If the model creates new classes, the task no longer reflects the developer’s actual refactoring decision. **Functional Tests** check that the move preserves program behavior instead of only making structural changes. **Executability** ensures that the resulting changes can be validated on a runnable project, which is essential for capturing dependency and import correctness in **Move Method**.

---

> ### Author Response · Authors · 2025-11-20
> **Authors' Rebuttal 3**
>
> **Table 2. Comparison of data-quality criteria for Move-Method instances in SWE-Refactor and RefactorBench.**
>
> | Dataset        | % Specified Move Target | % Existing Target | % with Functional Tests | % Executability     |
> |----------------|------------------------|-------------------|-------------------------|---------------------|
> | **SWE-Refactor**   | 100% (410/410)          | 100% (410/410)      | 100% (410/410)            | 100% (410/410)        |
> | **RefactorBench** | 0% (0/26)               | 62% (16/26)         | 0% (0/26)                 | not guaranteed        |
>
> **Specified Move Target.** For all 26 instances in RefactorBench, the input instructions already provide the ground truth on 1) which method to move and 2) where to move the method to. In other words, no opportunity identification is required, unlike SWE-Refactor, where the model must reason about and construct the behavior-preserving transformation itself.
>
> **Existing Target.** In SWE-Refactor, all target classes exist in the repository (100%). In RefactorBench, 38% of Move-Method instances target classes that do not exist (only 62% operate on real classes). When the target class is missing, the model must create a new class, which no longer reflects a real Move Method scenario and makes evaluation unclear.
>
> **Functional Tests.** Upon manual checks, all test cases in RefactorBench DO NOT verify whether the refactoring exists or is functionally correct. The test cases are limited to static structural checks, such as whether the target file contains the name of the moved method or whether an import statement has been updated.
>
> **Executability.** All instances in SWE-Refactor are runnable (100%), so we can check whether the move breaks the build or dependencies. By contrast, RefactorBench is not guaranteed to run, so build failures cannot be checked.
>
> [1] Tsantalis, Nikolaos, Ameya Ketkar, and Danny Dig. "RefactoringMiner 2.0." IEEE Transactions on Software Engineering 48.3 (2020): 930-950.
>
> [2] Silva, Danilo, et al. "Refdiff 2.0: A multi-language refactoring detection tool." IEEE Transactions on Software Engineering 47.12 (2020): 2786-2802.
>
> [3] PMD. Pmd - source code analyze, 2025. URL https://pmd.github.io/.
>
> [4] Cao, Jialun, et al. "Javabench: A benchmark of object-oriented code generation for evaluating large language models." Proceedings of the 39th IEEE/ACM International Conference on Automated Software Engineering. 2024.
>
> [5] Gautam, Dhruv, et al. "Refactorbench: Evaluating stateful reasoning in language agents through code." arXiv preprint arXiv:2503.07832 (2025).
>
> [6] Liu, Bo, et al. "Exploring the potential of general purpose LLMs in automated software refactoring: an empirical study." Automated Software Engineering 32.1 (2025): 26.
>
> [7] Test cases for new-inventory-patterns-task, https://github.com/microsoft/RefactorBench/blob/main/tests/ansible_refactor/new-inventory-patterns-test.py

---

### Author Response · Authors · 2025-12-03
**General Response to Main Questions**

We sincerely thank all reviewers for their constructive feedback.

Specifically, we appreciate the reviewers' recognition of:
* the originality of SWE-Refactor and the clear gap it fills in current benchmarks (`Rev. GqSu`, `Rev. 5ytC`);
* the clean, automated data-construction pipeline and high-quality, noise-free refactorings (`Rev. GqSu`, `Rev. aF72`, `Rev. 5ytC`);
* the comprehensive evaluation and careful analysis of multi-agent refactoring performance (`Rev. GqSu`, `Rev. aF72`);
* the clear writing and organization of the paper (`Rev. aF72`, `Rev. 5ytC`).

We summarize the main questions and our responses below.

**Mian Question 1: Are state-of-the-art models (e.g. GPT-4o, Claude 4.x Sonnet, Gemini 2.5 Pro) and agentic scaffolding able to “solve” the benchmark? (`Rev. 5ytC`)**

**Response:**
To check whether SWE-Refactor is already solved for the SOTA models, we added experiments with model  **GPT-4o**  and agentic scaffolding **GPT-5.1-Codex (agent mode)**  :
- **GPT-4o** solves **675/1,099 (61.4%)**, an improvement over **52.7%** with GPT-4o-mini.
- **GPT-5.1-Codex (agent mode)** solves **151/200 (75.5%)** in the sampled cases, close to its **74.5%** result on SWE-bench-Verified released by OpenAI.

These results show that SWE-Refactor remains challenging, especially for compound refactorings (e.g., solved 42% for Extract and Move). We also updated the paper to include these results.

**Main Question 2: Why does the dataset end up with 1,099 instances? What limits the scale, and what is the purity–scale trade-off? (`Rev. 5ytC`)**

**Response:**
The bottleneck is not the purity filter but the later quality checks. From **29,754** detected refactorings:
- **16,482** remain after purity filtering,
- **7,511** remain after validating target-file existence,
- **3,711** build and pass tests,
- **1,099** have test coverage.

Thus, buildability, test stability, and coverage checks determine the final scale.

**Main Question 3: If IDEs already support these refactorings, what is the practical value of having LLMs perform them? (`Rev. 5ytC`)**

**Response:**
Modern IDEs require manual decisions and guidance throughout every step. They do **not** decide:
- which block to extract,
- how to name the method,
- which parameters are needed,
- or which class a method should be moved to.

In contrast, SWE-Refactor evaluates the **full decision-making process**: finding the opportunity, choosing where and how to change the code, applying the refactoring, and checking it by compiling and running tests.

**Main Question 4: How is ground truth correctness ensured, and how does it compare to prior datasets? (`Rev. GqSu`)**

**Response:**
We ensure correctness using multiple checks:
- RefactoringMiner for detection,
- PurityChecker to ensure no behavioral edits,
- compilation and test execution,
- and coverage to confirm the refactored method is exercised.

We also compared our data with RefactorBench (only 100 instances and the tests only check if certain strings like package name exists) and found that SWE-Refactor provides more reliable targets, functional tests, and guaranteed executability.

**Main Question 5: Is the benchmark representative, given its language choice and the distribution of refactoring types? (`Rev. GqSu`, `Rev. aF72`)**

**Response:**
More than **95%** of existing SE benchmarks focus on Python. SWE-Refactor adds needed diversity by providing a large Java benchmark with **1,099** validated refactorings, including **177 compound refactorings (16.1%)**, which match their natural real-world frequency.
We also explained which pipeline components would need replacement to support other languages.

**Main Question 6: How efficient is the multi-agent workflow, and why does it work? (`Rev. aF72`)**

**Response:**

We added efficiency numbers:
- the agent uses **around 9.35** iterations per instance,
- and about **151K tokens** per run (~\$0.03 with gpt-4o-mini).

From a manual check of 50 examples, **32** refactorings were completed successfully:
**15** on the first try, and **17** after one or more rounds of reviewer–developer feedback.
These results show how the feedback loop helps the workflow correct earlier mistakes.

---

### Meta-Review · Area_Chair_4HTA · 2026-01-04

**Summary:**

The paper presents a new benchmark dataset, named SWE-Refactor, that evaluates LLM’s ability to perform code refactoring. It was collected from 18 real-world java projects, and synthetically generated. The benchmark is evaluated on a suite of LLMs (e.g., CodeLLaMA, GPT-4o-mini, DeepSeek-V3), showing they are limited in its refactoring ability.
The paper is well-written, and data generation process is sound and carefully documented. The results are comprehensive with analysis of multi-agent refactoring performances. This is a solid work. The work distinguishes itself from existing code benchmarks (e.g., repository-level refactoring with rich dependencies, their focus on java rather than python). One of the common reviewer concerns was limited scope (only studying Java, and this is fundamental to their data construction approach which relies on Java-specific tools). The authors have provided detailed comparison with existing benchmarks (SWE-Refactor and RefactorBench), showing the benefits of their benchmarks (e.g., better evaluation, more coverage of refactoring types). The reviewers asked of comparison of SoTa method, which the authors provided during the rebuttal. The new result shows 75.5% success rate for GPT-5-1-Codex, significantly outperforming systems presented in the original manuscript (41.6% success rate). The authors argue this is similar to SWE-Refactor and that is acceptable, but the AC shares the reviewer 5ytC’s concern that this dataset would be solved very soon. The AC is a bit hesitant to recommend acceptance given the limited scope, and the potential significance of the work in the community.

**Reviewer Concerns:**

See meta review.

**Reviewer Scores:**

The authors engaged seriously with the reviewers and provided thoughtful responses to each of them. So I could imagine reviewers having raised scores slightly. Having said that, I do not think the responses fundamentally would have changed the reviewer's actual opinion of the paper as there weren't really misconceptions or new positive results presented.

---

### Decision · Program_Chairs · 2026-01-26

Reject